# Predicting Dynamic Properties of Asphalt Mastic Considering Asphalt–Filler Interaction Based on 2S2P1D Model

**DOI:** 10.3390/ma15165688

**Published:** 2022-08-18

**Authors:** Xiaoyan Ma, Xingyu Zhang, Junpeng Hou, Shanglin Song, Huaxin Chen, Dongliang Kuang

**Affiliations:** 1Engineering Research Center of Transportation Materials, Ministry of Education School of Materials Science and Engineering, Chang’an University, Xi’an 710064, China; 2Research and Development Center of Transport Industry of Technologies, Materials and Equipments of Highway Construction and Maintenance, Gansu Road & Bridge Construction Group Co., Ltd., Lanzhou 730030, China; 3Scientific Observation and Research Base of Transport Industry of Long Term Performance of Highway Infrastructure in Northwest Cold and Arid Regions, Lanzhou 736200, China; 4National Center for Materials Service Safety, University of Science and Technology Beijing, Beijing 100083, China

**Keywords:** asphalt mastic, asphalt–filler interaction, 2S2P1D model, dynamic properties prediction

## Abstract

The relationship between the various phases of asphalt materials, from asphalt binder to mastic and mixture, has received great attention over the years, with efforts being made to establish linkages among these phases. Many methods for predicting the rheology properties of asphalt mastics from those of asphalt and filler volume fractions exist. However, most prediction methods are based on an empirical formula and on the micromechanical model. Very few research studies focus on the constitutive model. In addition, relatively little research has explored the influence of asphalt–filler interaction on mastic’s rheology properties, which is believed to be an important factor. In this study, the 2S2P1D (two springs, two parabolic elements, and one dashpot) model was applied to link the behavior of asphalt binder, filler volume fraction, asphalt–filler interaction and asphalt mastic. First, the interaction between asphalt and filler was evaluated, and the interaction parameter C of the Palierne model was used as an assessment indicator to calculate the effective filler volume fraction of asphalt mastic. Then, the relation between the 2S2P1D model parameters of asphalt mastic and those of asphalt binder and the effective filler volume fraction was analyzed. Finally, a simple relationship associating the 2S2P1D model parameters h, $\log(\tau_{0})$ of mastic and that of asphalt binder and the effective filler volume fraction was developed. The proposed expression was validated, and the result showed that it was an efficient model for the shear complex modulus prediction of virgin asphalt mastic.

## 1. Introduction

The relation among various phases of asphalt material is a hot topic and has garnered much interest over the years. Many researchers have invested a great deal of effort to propose or develop effective models for estimating the rheological properties of asphalt mixture compared with those of asphalt binder. Traditionally, the estimation of the effective properties of asphalt mixture is based on mixture scale experiments or only-asphalt–binder experiments. The drawback of the prediction mixture property from only asphalt is that this material consists of just 4–6% of a mixture by weight, and the interaction between asphalt and aggregate is not fully understood. In addition, using experiments to obtain the rheological properties of a mixture is time consuming, and when modeling a mixture from binder, one must assume the homogeneity of the composite to rigorously solve the mathematical equations [1]. As early as the 1930s, researchers recognized the significant influence of mastic on a mixture and conducted an extensive study to understand the effect of mineral filler on the properties of mastic [2]. These studies came to the general consensus that the best way in which to determine the effect of filler or asphalt on mixture is to experiment directly with the blended mastic. Asphalt mastic, which consists of asphalt binder and mineral filler with a diameter less than 0.075 mm, is the real binder for combining an aggregate of different sized particles together in a mixture. Therefore, a large number of researchers have attempted to understand the stiffness of mineral filler and asphalt with the goal of providing a bridge between asphalt and mixture [3,4].

Prediction modeling work has been largely developed, based on the relation between asphalt and mastic, and many regression models have been proposed. Yan described the relationship between the complex modulus, creep stiffness, and the filler/asphalt ratio as an exponential function [5]. Buttlar also proposed a simple exponential model for estimating the modulus of mastic from that of asphalt binder and the filler volume fraction [6]. In addition, with the benefit of the finite element method, Tehrani applied the two-dimensional (2D) and three-dimensional (3D) models to predict the complex modulus of asphalt mastics. He found that the calculated results were situated under experimental results [7]. However, due to the small number of asphalt mastic samples, the application of these models to the prediction mastic modulus is limited. Therefore, other researchers introduced micromechanical models from the field of matrix–particle composite materials. Based on application simplification and hypothesis, many micromechanical models for the prediction of the mastic modulus were developed. Yin employed the dilute model, self-consistent model, Mori–Tanaka model, and generalized self-consistent model to predict the complex modulus of mastic within the frequency of 0.01 to 10 at three controlled temperatures [8].

The proposal of these models provided methods and new ways of simulating the effective properties of asphalt mastic. However, the relation between asphalt and the mastic complex shear modulus in the above works was measured under the same condition of controlled point temperatures and frequencies, and the prediction models were only valid over a rather narrow range of temperatures and frequencies. Given these prediction methods’ shortcomings, Al-Khateeb proposed a simple regression exponential model to calculate the complex modulus master curve of mastic [2]. Kim selected the Nielsen model to predict the stiffening effect of the low filler volume fraction [9]. Hajikarimi regarded asphalt mastic as a heterogeneous medium consisting of aggregate particles, and he used the biphasic finite-element method to simulate asphalt mastic with a filler ratio of 0.18 and 0.35. This method proved to be successful for simulating the linear viscoelastic properties of mastic with a filler ratio of 0.18, but a significant difference was found between the numerical result and the observations of mastic with a filler ratio of 0.35 [10]. Pei employed the particle interaction model to predict the mastic modulus and found that it was suitable for mastic with a filler volume fraction not more than 0.50. Still, the determination of the parameter related to the Young modulus of mastic was not clear, which induced great uncertainty in the prediction process [11].

The application of these microstructure models in mastic modulus prediction helped with overcoming the drawbacks of models that validate only at fix conditions. This enabled the prediction process to move forward with a wide range of frequencies and temperatures. Nevertheless, the results varied, and these models were considered to possess better predictions for mastics with a filler volume fraction less than 0.25, which approaches the lower limit of the actual filler volume fraction (from 0.23 to 0.60) in practice.

To further improve the prediction accuracy, some researchers took the “structure asphalt” and the thickness of this rigid layer into account, as more and more research studies have confirmed the existence of this rigid layer [12,13,14,15]. Although the physical–chemical interaction between asphalt and mineral filler is not completely understood, any prediction of the effective properties of asphalt mastic must be tempered with the fact that an asphalt–filler interaction exists, and the interaction must significantly influence the observed behavior of asphalt mastic. Buttlar employed the physiochemical reinforcement into the prediction of the mastic modulus and replaced the filler volume fraction in the GSCS model with the effective filler volume fraction, which was the sum of the real filler volume fraction and the apparent immobilized asphalt volume fraction [6]. Underwood regarded the rigid asphalt layer as a new phase and used a four- phase GSCS model. This physical–chemical model is designed to calculate the modulus of asphalt mastic according to the modulus of asphalt binder, the filler volume fraction, and the thickness of the rigid layer [16]. These two studies largely improved the accuracy of prediction; however, they could predict the stiffening responses only at low and moderate filler volume fractions. They were unable to estimate the stiffening for mastic with a filler volume fraction more than 0.50. What is more, in these studies, the prediction of the extent of filler stiffening on asphalt in terms of rheological property changes ignored the fact that mineral filler greatly changes not only the value of the complex modulus but also the phase angle of asphalt, and few existing works in the literature involve phase angle prediction.

In addition to the fitting models and microstructure models, a more successful approach based on the 2S2P1D (two springs, two parabolic elements, and one dashpot) model together with the Shift-Homothety-Shift in time-Shift transformation (SHStS) was applied to predict the effect properties of asphalt mastic and a mixture [17,18,19]. With three geometrical transformations (a negative translation along the horizontal axis, a homothetic expansion, and a positive shift along the real axis), the modulus of asphalt mastic, fine asphalt mixture, and asphalt concrete could be calculated according to the modulus of asphalt binder at the same frequency and temperature. Riccardi considered the SHSTS transformation to be an effective tool for predicting the modulus and phase angle of asphalt mastics compared with those of asphalt binder by simply adding a transformation parameter, which was relevant only for the filler volume fraction [19]. According to his method, the prediction of the asphalt mastic complex modulus is possible if the modulus of binder, the minimum modulus and maximum modulus of mastic, and the parameters related to the statistical spatial distribution of mineral filler in mastic are known. However, the determination of the transformation parameters is not clear, and the physical–chemical interaction was not involved in this study.

Although several research studies have been carried out on asphalt mastic modulus prediction, the performance of mastic is not fully recognized, and a sufficient characterization of the stiffening effect of filler on asphalt is certainly worth the effort. Providing an accurate as well as a practical model for the estimation of asphalt mastic effective properties will help researchers to reasonably select asphalt and filler as well as obtain approximate formulas that can be applied in the mixture design for the purpose of generating the first estimates. It also makes a larger-scale asphalt material prediction phase, such as that for a fine asphalt mixture or asphalt concrete, more reliable.

## 2. Materials and Tests

### 2.1. Materials

In the present research study, three virgin asphalt binders and two mineral fillers were employed to prepare mastic samples. Three asphalt binders were produced with different sources of crude oil and via different oil companies. They were named SK90, KL70 and ZH70 according to their manufacturers and penetration grades. The performance of the three asphalt binders is listed in Table 1. Two limestone powders (named A and B), each with a diameter less than 0.075 mm, were selected as the mineral fillers to fabricate 14 asphalt mastics with four 4 filler volume fractions (0.23, 0.38, 0.53, and 0.68 by volume of the total amount of asphalt binder and filler). The densities of the two fillers were 2.685 g/cm^3^ (A) and 2.749 g/cm^3^ (B), respectively. The specific surface areas were 482 m^2^/kg (A) and 495 m^2^/kg (B), respectively.

The fabrication of asphalt mastic was carried out with the help of a small mixer in a laboratory. Before mixing, asphalt binder and mineral filler were put in an oven at 135 °C to melt the asphalt binder completely and to heat the mineral filler to the appreciation temperature. Then, the heated filler was gradually added to asphalt while stirring for at least 20 min until the mix was evenly blended. At last, the mix was pulled into silicone molds with a diameter of 8 mm or 20 mm to test its rheological property.

### 2.2. Tests

An AR2000 dynamic shear rheometer (DSR) by TA (Milford, MA, USA). Corporation was employed to obtain the rheological performance of asphalt binder and mastics. Two types of silicone molds with a diameter of 8 mm and a height of 2 mm—or a diameter of 20 mm and a height of 1 mm—were selected for asphalt and mastic molding. According to the principle that a mold with a small diameter and a great height applies to high-modulus materials, and vice versa, the 8 mm mold was used to cast asphalt and mastics that would be tested at temperature lower than 30 °C, and the 20 mm mold was used to cast asphalt materials tested at a temperature higher than 30 °C.

To ensure that the rheological properties of asphalt and mastics were tested within the linear viscoelasticity limit (LVE), strain sweep tests were conducted to obtain the shear strain of the frequency sweep. Then, based on the LVE, the frequency sweep tests were conducted at a temperature from 15 to 70 °C and at a frequency from 0.01 to 100 rad/s to obtain the complex modulus and phase angle of the asphalt and mastics. Three replicates were conducted for each asphalt and mastic. The average value was calculated to obtain its complex modulus and phase angle.

## 3. 2S2P1D Model

2S2P1D is a rheological model derived from the Huet–Sayegh model by adding a linear dashpot to the two parabolic and a series of springs (Figure 1). Since École Nationale des Travaux Publics de l’État (ENTPE) proposed it in France in 2003, the 2S2P1D model has been applied to fitting the complex modulus and phase angle master curves of asphalt binder, asphalt mastics, and mixtures. This model overcomes the drawbacks of the analytical expression (such as the Sigmodel model, Christensen–Anderson model, and Christensen–Anderson–Marasteanu model) of the complex modulus and phase angle master curves. It establishes a linkage between the rheological properties of asphalt materials and mechanics [17].

The expression of the 2S2P1D model for fitting the master curve of the modulus of asphalt at a reference temperature is given as follows:(1)$$G_{}^{*}(i\omega \tau )=G_{0}^{*}+\frac{G_{\infty }^{*}-G_{0}^{*}}{1+\delta {\left(i\omega \tau \right)}^{-k}+{\left(i\omega \tau \right)}^{-h}+{\left(i\omega \beta \tau \right)}^{-1}}$$
where $G^{*}$ is the complex modulus; $G_{0}^{*}$ and $G_{\infty }^{*}$ represent the static modulus (ω→0) and glassy modulus (ω→∞), respectively; ω is the angular frequency of shear, and $\omega =2\pi f_{r}$; $f_{r}$ is the reduced frequency; i is the complex number defined by $i^{2}=-1$; τ is the characteristic time, which represents the temperature dependency of the corresponding asphalt material; k and h are exponents, defined as 0<k<h<1; and β is a constant, which is related to Newtonian viscosity η, where $\eta =\beta (G_{\infty }^{*}-G_{0}^{*})$.

More details on the parameters can be expressed in the Cole–Cole diagram (Figure 2). It shows that h and k govern the slope of the diagram at low and high values of the storage modulus (real $G^{*}$); *δ* is related to the height of the maximum point in the Cole–Cole diagram.

## 4. Result and Discussion

### 4.1. Test Result

As the Black diagram (Figure 3) indicates, none of the three asphalts obeys the time–temperature superposition principle, as their complex modulus–phase angle curves are not unique. Therefore, the complex modulus and phase angle master curves of the asphalt binders and the corresponding mastics were obtained by shifting the plots to a reference temperature based on the least square method. Figure 4, Figure 5 and Figure 6 show the complex modulus and phase angle master curves at 40 °C for the asphalt and mastics obtained via this shifting method.

The master curves of the complex modulus and phase angle for the three asphalt binders and their corresponding mastics (Figure 4, Figure 5 and Figure 6) showed that the filler volume fraction has a very significant influence on asphalt stiffness at a low frequency (or a high temperature), and the complex modulus increases with the filler volume fraction. However, at a very high frequency, the master curves for different filler volume fractions remain close to one another, and the influence of filler on asphalt stiffness gradually weakens. For example, with the filler volume fraction increases from 0.23 to 0.68, the increasing complex modulus folds are 2.23-, 3.47-, 4.55-, and 7.26-fold, respectively, with a frequency of 0.00661 Hz. The increasingly complex modulus folds are 2.15-, 3.16-, 4.03-, and 5.70-fold, respectively, at a frequency of 59.75 Hz.

Not only does the loading frequency influence the filler volume fraction on the complex modulus but also the constituent materials of asphalt binder and mineral filler do as well. For the asphalt mastics with a filler (A) volume fraction from 0.23 to 0.68, the increasing complex modulus folds are 2.69-, 4.11-, 5.04-, and 8.44-fold for SK90 asphalt binder at a frequency of 0.15 Hz. However, the increasing folds are 2.50-, 3.89-, 5.30-, and 7.42-fold for KL70 asphalt binder at a frequency of 0.15 Hz. For mastics with the same binder (KL70) and for different types of fillers with the same volume fraction (0.23), the increasing complex modulus folds are 2.50- and 1.95-fold for filler A and B, respectively.

As shown in Figure 4b, Figure 5b and Figure 6b, the phase angle approaches its maximum value of about 90°, which indicates that the matrix asphalt binder exhibits a purely viscous behavior at very high temperature or low frequency. Compared to the appreciable effect of mineral filler on complex modulus of asphalt binder, the impact of filler to the phase angle is rather small.

### 4.2. The Interaction between Asphalt and Filler

As mentioned in the previous paragraph, it is clear that the rheological properties of asphalt mastics affected the filler volume fraction. It is worth noting that asphalt mastics were prepared with the same filler volume fraction, the same asphalt binder and different types of mineral filler possessing unequally complex modulus. In addition, asphalt mastics were prepared with the same filler volume fraction, mineral filler, and different types of asphalt binders. So, it is reasonable to believe that such a discrepancy is related to the asphalt–filler interaction. The physical–chemical interactions occurring between asphalt and filler caused some of the asphalt to be absorbed on the surface filler and form a rigid layer. The existence of this layer makes the actual volume of solid particles much greater than the calculated volume of particles (filler) by mass and density. The rate of particle volume growth is dependent on this layer’s thickness. In addition, the thickness is determined based on the physical–chemical interaction between the asphalt and filler. Research on the relation of film thickness and the filler volume fraction showed a maximum adsorption potential equal to 0.60. However, as the filler volume fraction grows above 40%, the film thickness must drop, as only 30% of the total asphalt volume could be adsorbed on the filler surface [16]. The curve of the film thickness with the filler volume fraction indicates that the fraction of the adsorbed rigid asphalt varies. This is because the amount of filler volume and the film thickness are related to physical parameters, such as a specific surface area of filler. This understanding helps with advancing the exploration of the adsorbed rigid layer. However, it cannot characterize the degree of interaction stemming from the asphalt performance and filler chemical composition. To quantitatively evaluate the interaction between asphalt and filler, researchers proposed some indicators based on the rheology of asphalt and mastics, such as the complex modulus coefficient, ΔG*; the complex viscosity coefficient, $\Delta \eta^{*}$; the asphalt–filler interaction evaluation index, B (K−B−G*), which is derived from the Palierne model; the interface energy loss parameter, *A* (L−A−δ), which is derived from the three- phase model; the composite materials mechanical damping coefficient, $\tan\delta_{c}$; and, the Einstein coefficient, K_E_, of the Nielsen model. A reasonable asphalt–filler interaction evaluation index should be closely related to the thickness of the rigid layer absorbed on the filler surface. This implies that the interaction evaluation index is independent of the filler concentration when the filler volume fraction is below the critical volume fraction. Among these interaction evaluation indicators, only the interaction parameter C of the Palierne model is consistent with the above criteria [8]. The Palierne emulsion model can be written as follows:(2)$$G_{c}^{*}(\omega )=G_{m}^{*}(\omega )\frac{1+3\sum {}_{i}\phi_{i}H_{i}(\omega )}{1-2\sum {}_{i}\phi_{i}H_{i}(\omega )}$$
(3)$$H_{i}(\omega )=\frac{4(\alpha /R_{i})\left[2G_{m}^{*}(\omega )+5G_{i}^{*}(\omega )\right]+\left[G_{i}^{*}(\omega )-G_{m}^{*}(\omega )\right]\left[16G_{m}^{*}(\omega )+19G_{i}^{*}(\omega )\right]}{40(\alpha /R_{i})\left[G_{i}^{*}(\omega )+G_{m}^{*}(\omega )\right]+\left[2G_{i}^{*}(\omega )+3G_{m}^{*}(\omega )\right]\left[16G_{m}^{*}(\omega )+19G_{i}^{*}(\omega )\right]}$$
where $G_{i}^{*}(\omega )$ is the complex modulus of the dispersed phase; $G_{m}^{*}(\omega )$ is the complex modulus of dispersing media; $G_{c}^{*}(\omega )$ is the complex modulus of the composite material; and $\phi_{i}$ is the dispersed phase volume fraction with a radius of $R_{i}$.

As the radius of each dispersed phase can be replaced with the mean radius of all of the dispersed phases, then the Palierne model can be written as:(4)$$G_{c}^{*}(\omega )=G_{m}^{*}(\omega )\frac{1+3\phi H(\omega )}{1-2\phi H(\omega )}$$
(5)$$H_{i}(\omega )=\frac{4(\alpha /R_{i})\left[2G_{m}^{*}(\omega )+5G_{i}^{*}(\omega )\right]+\left[G_{i}^{*}(\omega )-G_{m}^{*}(\omega )\right]\left[16G_{m}^{*}(\omega )+19G_{i}^{*}(\omega )\right]}{40(\alpha /R_{i})\left[G_{i}^{*}(\omega )+G_{m}^{*}(\omega )\right]+\left[2G_{i}^{*}(\omega )+3G_{m}^{*}(\omega )\right]\left[16G_{m}^{*}(\omega )+19G_{i}^{*}(\omega )\right]}$$

For the mineral particles dispersed in asphalt binder, $G_{i}^{*}(\omega )\to \infty$, and H(ω) equals 0.5. Thus, the Palierne model can be expressed as:(6)$$G^{*}(\omega )=G_{m}^{*}(\omega )\frac{1+1.5\phi }{1-\phi }$$

When the dissipation energy is considered, filler volume fraction ϕ should be replaced with $\phi_{F}C$, where $\phi_{F}$ is the volume fraction of filler particles, and *C* represents the filler–asphalt interaction. This interaction is related to the asphalt properties, the filler particle specific surface area, the density, structure, and the mineral composition. Then, Equations (7) and (8) can be expressed as follows, and $\phi_{F}C$ can be considered to be the volume of filler and adsorbed asphalt on the surface.
(7)$$G_{c}^{*}(\omega )=G_{m}^{*}(\omega )\frac{1+1.5\phi_{F}C}{1-\phi_{F}C}$$
(8)$$C=\frac{G_{c}^{*}(\omega )/G_{m}^{*}(\omega )-1}{\left[1.5+G_{c}^{*}(\omega )/G_{m}^{*}(\omega )\right]\phi_{F}}$$

Taking the calculation of the *C* value for the mastic manufacturing of KL70 and filler A, with a filler volume fraction of 0.23, for instance, the *C* value shows a slight fluctuation with the reduced frequency and the trend that the *C* value decreases with increased frequency. Therefore, data at a frequency of 0.0001 and 150 Hz are adopted to calculate the *C* value, and the average is taken as the final result. The C value of the 12 asphalt mastics are listed in Table 2. A weakened effect of the interaction between asphalt and filler with increases in the filler volume fraction can be clearly observed in Table 2, as indicated by changes in the *C* value. An explanation for this is that the thickness of the adsorbed asphalt decreased with the filler volume fraction. The amount of adsorbable asphalt was limited, and the increase in the of filler volume fraction meant a rise in the surface area, which required more asphalt. This resulted in a decrease in the adsorbed filler thickness with the filler volume fraction, and it was reflected in a decrease in *C*. According to Underwood and other researchers, the relation between the adsorbed film thickness and the particle concentration can be calculated [1,16]. It can be seen from Table 2 that great differences in the *C* value exist between mastics made of different types of asphalt and fillers. The *C* values indicate that filler (A) shows a more stronger interaction with SK90 asphalt than KL70 does, and KL70 asphalt shows a stronger interaction with filler (A) than with filler (B). This means that both asphalt and filler influence the interaction. With regard to the asphalt and filler selected in this research, it is obviously that the effect of asphalt on the filler–asphalt interaction is greater than that of the filler.

As mentioned previously, the effective filler volume fraction of $\varphi_{e}$ should be expressed by $\phi_{F}C$. So, in the following research, the relation between effective filler volume fraction $\varphi_{e}$ and parameters of 2S2P1D is analyzed.

### 4.3. Calibration of the 2S2P1D Model

#### 4.3.1. Determination of $G_{0}$ and $G_{\infty }$

At a reference temperature, the linear viscoelastic behavior of asphalt material could be entirely determined based on the seven constants ($G_{0}^{*}$, $G_{\infty }^{*}$, k, h, δ, τ and η) of the 2S2P1D model (Equation (1)). Ordinarily, the goodness of fit of the predicted model depends on the initial values of $G_{0}$ and $G_{\infty }$. Parameters $G_{0}$ and $G_{\infty }$ can be regarded as the static and glassy modulus of asphalt and mastics. For binders, the experimental static modulus is very close to zero. In addition, for asphalt mastics, $G_{0}^{*}$ can be predicted via the cluster–cluster aggregation (CCA) balance equation: $G_{0}^{*}=\underset{\omega \to 0}{\lim}G^{\prime }(\omega )$.

According to previous research studies of asphalt binder, when $G_{\infty }$ is between 50 and 200 MPa, the 2S1P1D model predicts well with the experimental data. Airey suggested using a value of 100 to 200 Mpa as the $G_{\infty }$ of the asphalt binder. Asphalt mastic can be regarded as a particles-filled polymer composite system [20,21]. On the basis of the principle of polymers that filler has reinforced, the dynamic mechanical parameters of filler-reinforced polymer are compatible with the time-concentration superposition (TCS). According to the first TCS, when the filler volume is less than the critical volume fraction, and when the frequency is more than 20 rad/s, the rheological behavior of the composite system is dependent on the polymer. The modulus curve of the composite system is parallel to that of the polymer, and the phase angle curve of the composite system is basically identical to that of the polymer. In this condition, the modulus shift factor ($A_{f}$) is introduced, and it is defined as follows:(9)$$A_{f}=\frac{G_{composite}^{*}}{G_{matrix}^{*}}\;\mathrm{or}\;A_{f}=\frac{G_{composite}^{\prime }}{G_{matrix}^{\prime }}$$

The complex modulus of a particle-filled polymer composite system can be expressed as follows:(10)$$G^{*}(\omega ,\varphi )=(1-\varphi )A_{f}(\varphi )G_{m}^{*}(\omega )+\varphi G_{f}^{*}(\omega ,\varphi )$$

As the complex modulus of filler dependencies on the frequency can be ignored, the equation can be expressed as
(11)$$G^{*}(\omega ,\varphi )=(1-\varphi )A_{f}(\varphi )G_{m}^{*}(\omega )+\varphi G_{f}^{}$$

In this study, the complex modulus master curve of mastics with different filler volume fractions is parallel to that of asphalt binder. In addition, the phase angle curve of asphalt mastic is basically identical to that of asphalt binder. Therefore, the $G_{\infty }^{*}$ of asphalt can be calculated by Equations (10) and (11). According to research studies on geotechnical mechanics, different rocks have an elastic modulus in the range of 1–100 GPa. In this study, the elastic modulus (E) of limestone is defined as 50 GPa. Based on the relation between different elastic constants, the shear modulus of limestone can be calculated as follows:(12)$$G_{f}=\frac{E}{2(1+\nu )}=\frac{50\mathrm{GPa}}{2\times (1+0.3)}=19.2\mathrm{GPa}$$

For asphalt mastics of different filler volume fractions, the 2S2P1D parameters of $G_{0}^{*}$ and $G_{\infty }^{*}$ can be estimated. Thus, the number of constants of the 2S2P1D model can be reduced to five. They could also be computed by minimizing the sum of the square of the distance between the tested complex shear modulus and that of the 2S2P1D model at N points in angular frequency $\omega_{i}$ [17,19,22]. The minimization is made at the reference temperature of 40 °C, using the Solve feature of MS Excel as Equation (13).
(13)$$\sum_{i=1}^{N}\left[{\left(G_{1}{}^{\mathrm{tested}}(\omega_{i})-G_{1}{}^{\mathrm{model}}(\omega_{i})\right)}^{2}+{\left(G_{2}{}^{\mathrm{tested}}(\omega_{i})-G_{2}{}^{\mathrm{model}}(\omega_{i})\right)}^{2}\right]$$
where $G_{1}{}^{\mathrm{tested}}(\omega_{i})$ and $G_{2}{}^{\mathrm{tested}}(\omega_{i})$ are the real part and the imaginary part of the tested complex shear modulus at angular frequency $\omega_{i}$; $G_{1}{}^{\mathrm{model}}(\omega_{i})$ and $G_{2}{}^{\mathrm{model}}(\omega_{i})$ are the real part and the imaginary part of the fitted complex shear modulus at angular frequency $\omega_{i}$ via the 2S2P1D model.

The parameters of the 2S2P1D model for all of the binders and corresponding mastics are shown in Table 3. The seven constants of the 2S2P1D showed that asphalt and its corresponding mastics with different filler volume fractions have the same value of δ, β, which means that these parameters depend only on the asphalt binder. The value of k, h, and $\log(\tau_{0})$ vary according to the filler’s volume fraction. For the parameters of k and h, their values decreased from 0.9649 to 0.9464; they also decreased from 0.9518 to 0.9311 for the SK90 and KL70 asphalt, respectively. Their corresponding mastics as the filler volume fraction increased, as h is a representation of the parabolic element, and its value expresses the slope of real $G^{*}$–image $G^{*}$ at a high temperature (low frequency). The decrease in h indicates that the ratio of real $G^{*}$ to image $G^{*}$ increased. The decrease in h with the filler volume fraction means an improvement in the elastic properties of asphalt [23]. In addition, the absolute value of parameter $\log(\tau_{0})$ also decreased from 3.3392 to 2.6547, and it decreased from 3.0624 to 2.4894 for SK90 and KL70 asphalt, respectively, and their corresponding mastics. As $\log(\tau_{0})$ is a function of temperature and accounts for the time–temperature superposition principle, a decrease in $\log(\tau_{0})$ means a decrease in the temperature sensitivity of the asphalt. It is worth noting that the value of parameter *k* defaults to *h*. As in Figure 2, k represents the slope of real $G^{*}$–image $G^{*}$ at a low temperature (high frequency) close to the glassy transition temperature. However, in this paper, due to a restricted operation of the DSR instrument, the lowest frequency sweep temperature is 20 °C, and this temperature is far from the glassy transition temperature of asphalt (about −20 °C). Therefore, the k value is the default to equal h.

Figure 7, Figure 8 and Figure 9 present the comparison between the tested complex shear modulus on asphalt binders and corresponding mastics and that predicted by using the 2S2P1D model. As shown in these figures, the model predictions fit the tested result reasonably well.

#### 4.3.2. Relationship between the 2S2P1D Model Parameters of Asphalt and Corresponding Mastics

When one is plotting the effective filler volume fraction of mastics (fabricated with different types of asphalt binder fillers), with the 2S2P1D parameter of h at 40 °C as shown in Figure 10, the relation of h and the filler volume fraction can be obtained. Figure 10a shows an apparent negative linear correlation between h and the filler volume fraction. When the filler volume fraction was replaced with the effective filler volume fraction, as shown in Figure 10b, curves were fitted, and parameters were obtained according to the relation between 
 and the filler volume fraction. When the filler volume fraction was replaced with the effective filler volume fraction, as shown in Figure 10b, curves were fitted, and parameters were obtained according to the relation between h of asphalt and its corresponding mastics (Table 4). The intercept of the fitting function was equal to the h of the asphalt binder, and the slopes of the three functions were −0.0320, −0.0332, and −0.0324, respectively. We could have h equal to −0.032, which means that the slope of the regression line is completely irrelevant to the asphalt binder. Then, the relation between the h of the asphalt binder and its corresponding asphalt mastic can be described by Equation (14):(14)$$h_{mastic}=h_{binder}-0.032\times \varphi_{e}$$

The plotting of the effective filler volume fraction of mastics and the 2S2P1D parameter of is shown in Figure 11. The trend of the characteristic time of asphalt mastic versus the different effective filler volume fraction could be obtained. Figure 11a shows an apparent positive linear correlation between $\log(\tau_{0})$ and the filler volume fraction. The $\log(\tau_{0})$ increased with the filler volume fraction. When we used the effective filler volume fraction in place of the filler volume fraction, as shown in Figure 11b, the line was fitted, and parameters were obtained according to the relation (Table 5). The intercept of the line was equal to the $\log(\tau_{0})$ of the asphalt binder, and the slopes of the three functions were 0.62132, 0.59193, and 0.59784, respectively. Concordance was obvious among the three functions. Therefore, we could have $\log(\tau_{0})$ equal to 0.60, which means that the slope of the regression line depends only on the effective filler volume fraction, and no link exists between $\log(\tau_{0})$ and asphalt binders. Therefore, the relation between the $\log(\tau_{0})$ of the asphalt binder and its corresponding asphalt mastic can be described by Equation (11):(15)$$\log\tau =\log\tau_{\mathrm{binder}}+0.60\varphi_{e}$$

### 4.4. Validation

Using the relation of the 2S2P1D parameters of the asphalt binder and its corresponding mastics described in the previous paragraph, the 2S2P1D parameters of the asphalt mastic could be obtained according to the given parameters of the asphalt and the effective filler volume fraction. For detecting the reliability of this prediction function, the third type of asphalt binder of ZH70 with a filler volume fraction of 0.38 to 0.53 was selected. In addition, the complex modulus and phase angle of the asphalt binder and mastics were measured. Then, the predicted complex modulus of the two asphalt mastics were compared with the experimental measurement results.

As previously described, the 2S2P1D parameters of *δ* and *β* are the same for asphalt and its corresponding mastics. The values of $G_{0}^{*}$ and $G_{\infty }^{*}$ of the asphalt mastic were calculated using the $G_{0}^{*}$ and $G_{\infty }^{*}$ of the asphalt binder as well as the modulus of the mineral filler and its volume fraction. The 2S2P1D parameters h and logτ of asphalt mastic were determined based on the asphalt binder and the effective filler volume fraction. Therefore, the relation between the complex modulus of the asphalt mastic and binder can be formulated as in Equation (16):(16)$$G_{mastic}^{*}(\omega ,T)=G_{}^{*}(G_{0}^{*}(\varphi ),G_{\infty }^{*}(\varphi ),h(\varphi_{e}),\log\tau (\varphi_{e}),\omega ,T)$$

Figure 12 and Figure 13 provide a comparison between the measured complex modulus of ZH70 (0.38A) and ZH70 (0.53A) with that of the predicted. The mean squared error of prediction in percentage form is equal to 4.9%; this suggests that the model predictions are very satisfactory.

## 5. Conclusions

In this paper, three asphalt binders and the corresponding mastics of various filler volume fractions were selected to investigate the possibility of setting up a simple model for predicting the complex modulus of asphalt mastic based on the 2S2P1D model. The shear complex modulus and phase angle of asphalt and mastics were obtained via a temperature-frequency sweep. Second, the seven parameters of the 2S2P1D model were calibrated for each asphalt and mastic based on the shear complex modulus and phase angle. Then, the relationship between the 2S2P1D parameters of asphalt binder and mastic was determined, and the forecast model was built. Finally, the reliability of the forecast model was validated.

Based on the performed analysis, the following conclusions can be drawn:

(1) The evaluation of the interaction between asphalt binder and filler showed that the type of asphalt binder, mineral filler, filler volume faction affected the interaction. In addition, the study revealed a weakened effect of the interaction between the asphalt and filler with increases in the filler volume fraction.

(2) For the seven parameters of the 2S2P1D model, the asphalt and corresponding mastics have the same value of δ and β. The $G_{0}^{*}$ and $G_{\infty }^{*}$ of the asphalt mastic could be calculated according to the time–temperature–concentration superposition principle based on the shear complex modulus of asphalt and mineral filler. The relationship between the h and $\log(\tau_{0})$ of asphalt binder and asphalt mastics follows a linear trend.

(3) The comparison result of the predicted complex modulus of asphalt mastics and experimentally measured data proved the reliability of the model.

The developed methodology for the shear complex modulus prediction of asphalt mastic was considered to be applicable to virgin asphalt binders and mineral filler. However, the 2S2P1D model parameters obtained in this paper is just a consequence of having rheological properties measured over a limited temperature range. It also has a limitation when it comes to predicting the shear complex modulus of polymer-modified asphalt mastic. For these unconventional materials, the fundamental assumption (i.e., time–temperature equivalence and phase angle-frequency relation) would produce unacceptable prediction errors. Special research effort is warranted in this respect.

## Figures and Tables

**Figure 1 materials-15-05688-f001:**
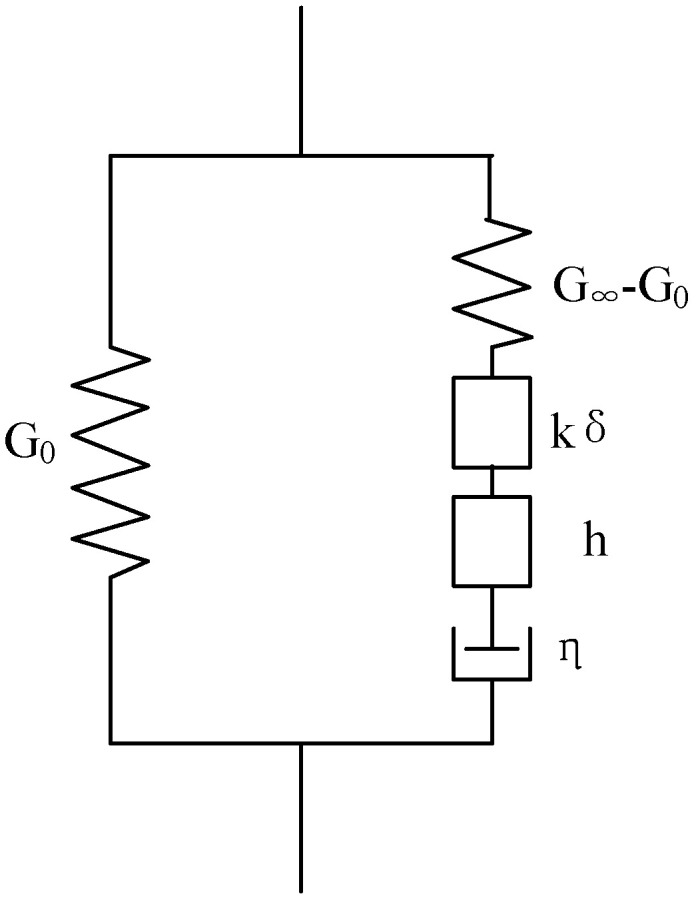
2S2P1D model.

**Figure 2 materials-15-05688-f002:**
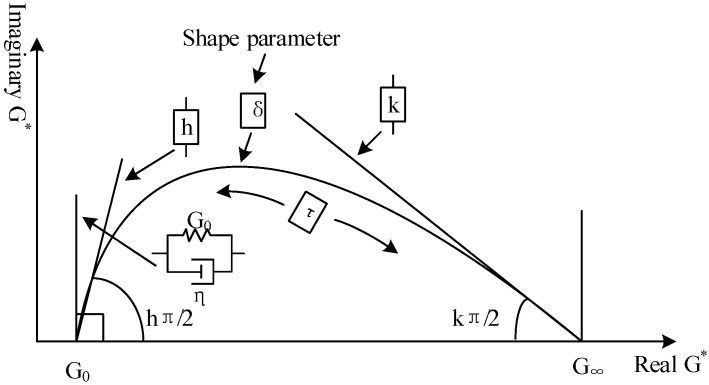
Visualization of the parameters of 2S2P1D model.

**Figure 3 materials-15-05688-f003:**
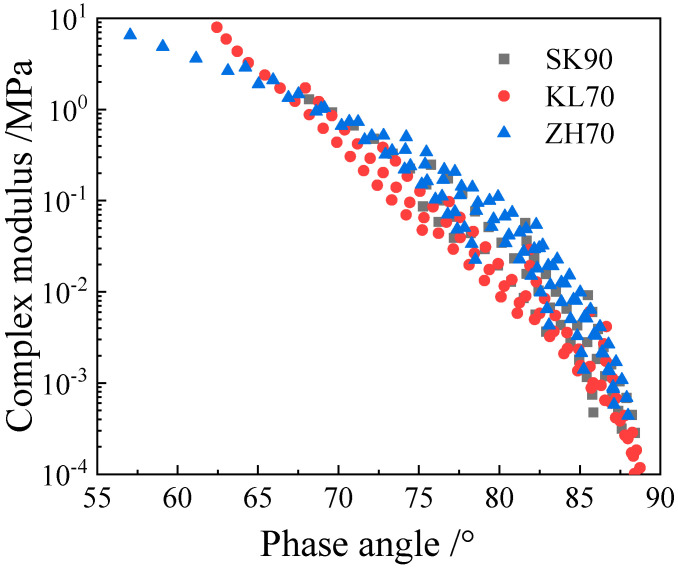
Black diagrams of the four asphalt binders.

**Figure 4 materials-15-05688-f004:**
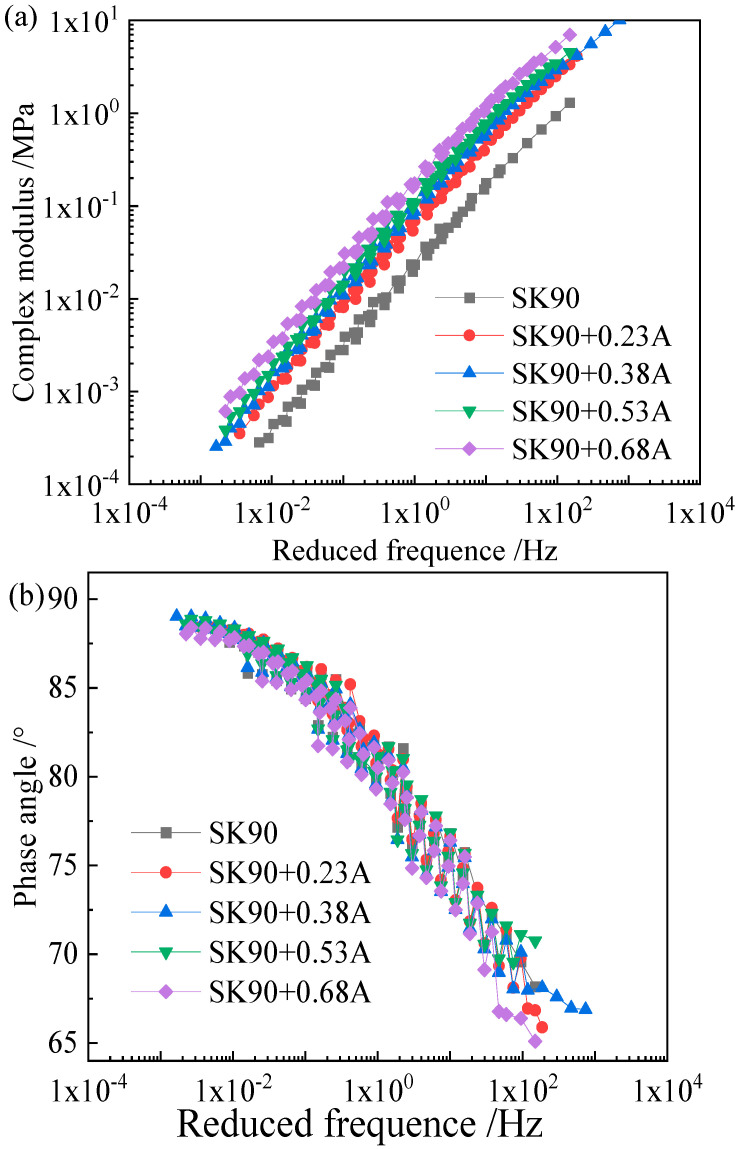
Master curve of complex modulus and phase angle of SK90 binder and mastics. (**a**) is the master curve of complex modulus of SK90 binder and mastics, (**b**) is the master curve of phase angle of SK90 binder and mastics.

**Figure 5 materials-15-05688-f005:**
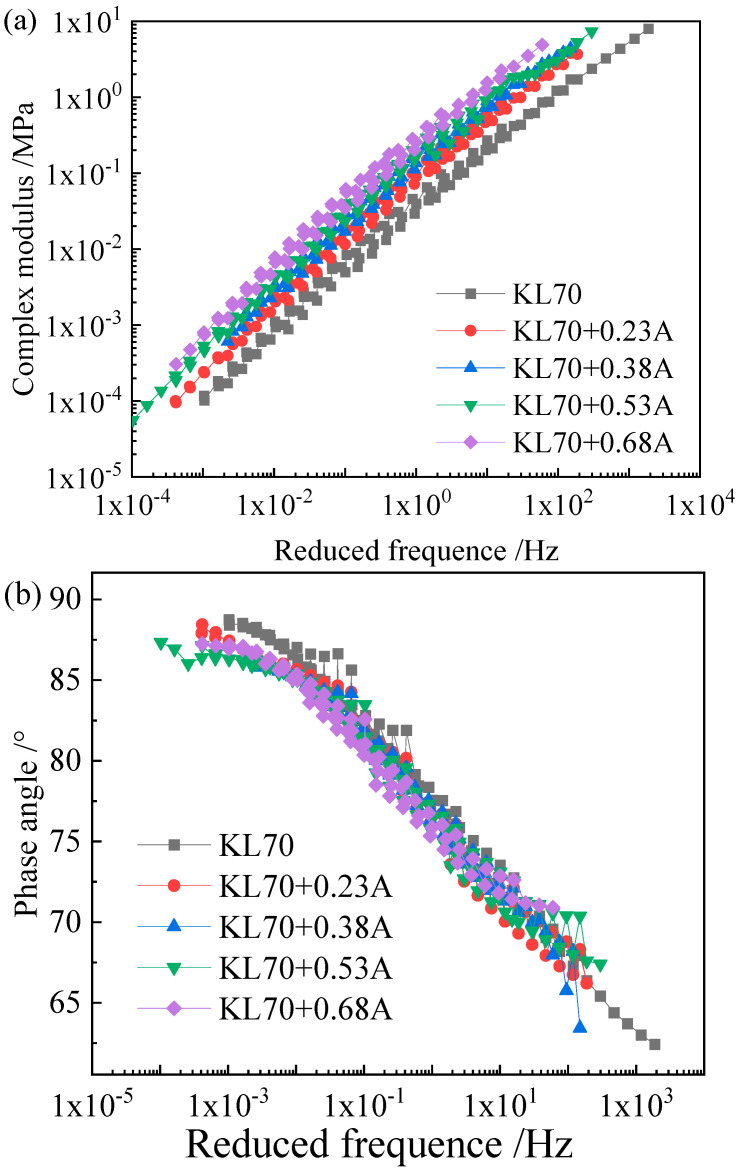
Master curve of complex modulus (**a**) and phase angle (**b**) of KL70 binder and mastics (A).

**Figure 6 materials-15-05688-f006:**
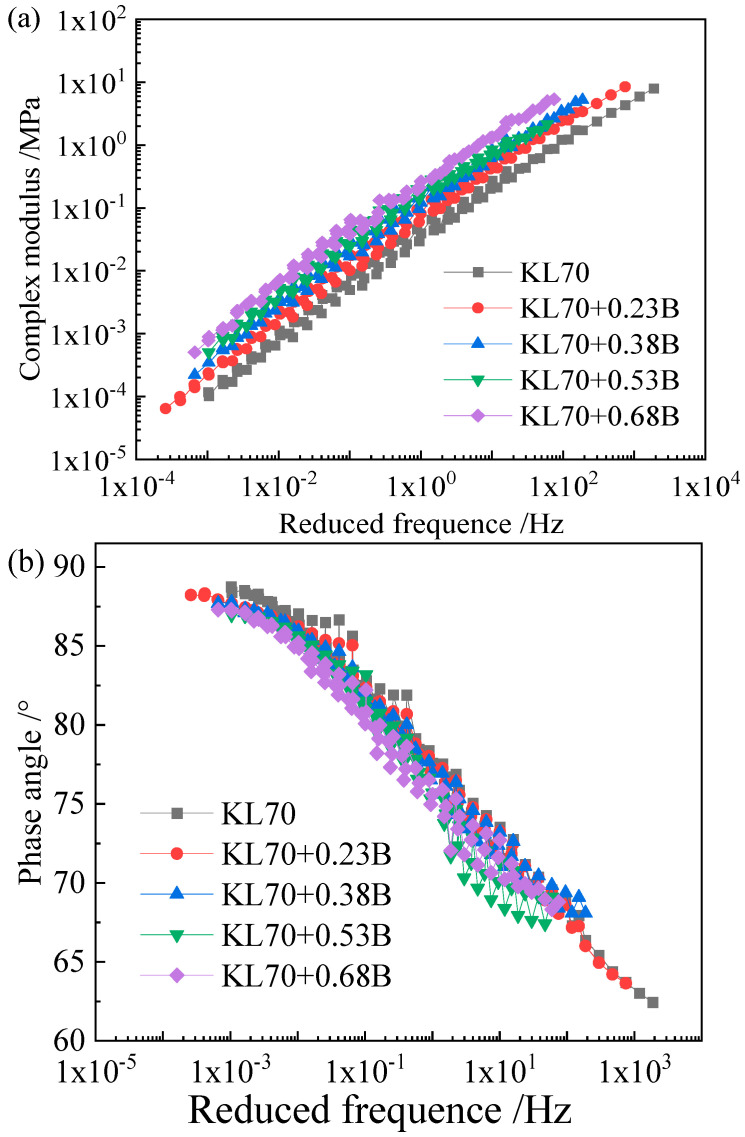
Master curve of complex modulus (**a**) and phase angle (**b**) of KL70 binder and mastics (B).

**Figure 7 materials-15-05688-f007:**
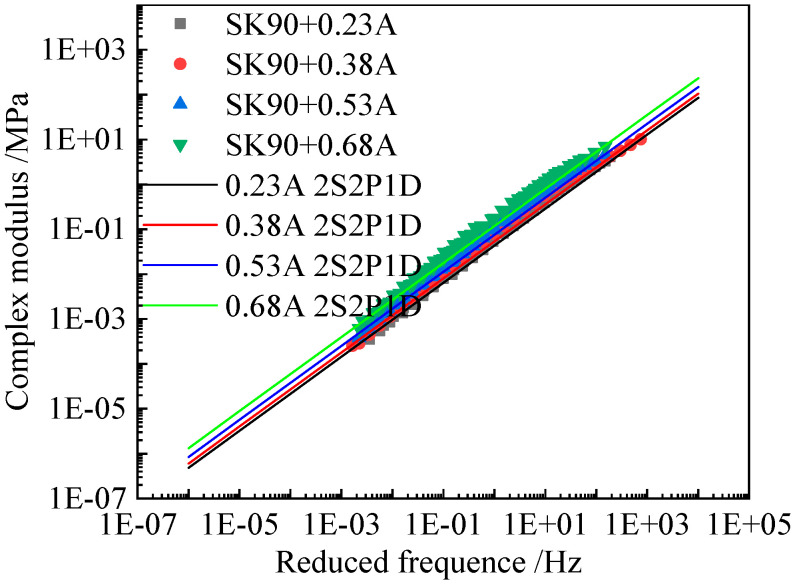
Comparison between the tested complex modulus master curves and 2S2P1D model fitted results of SK90 (A) asphalt mastics.

**Figure 8 materials-15-05688-f008:**
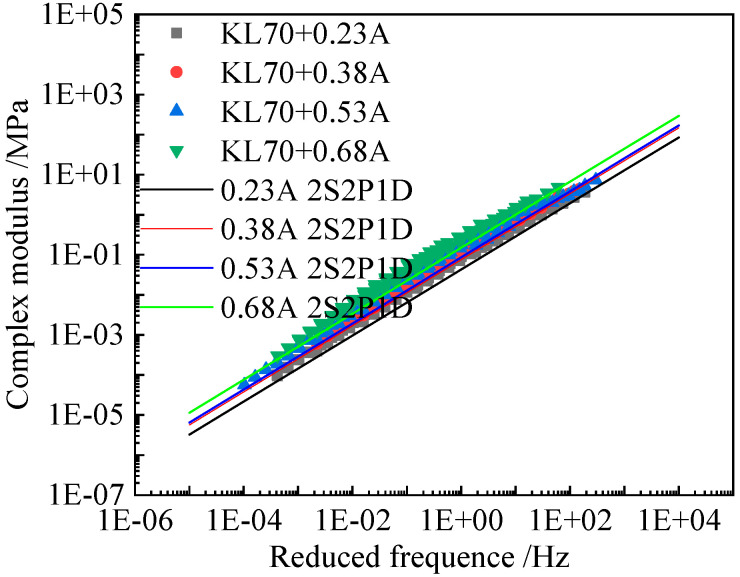
Comparison between the tested complex modulus master curves and 2S2P1D model fitted results of KL70 (A) asphalt mastics.

**Figure 9 materials-15-05688-f009:**
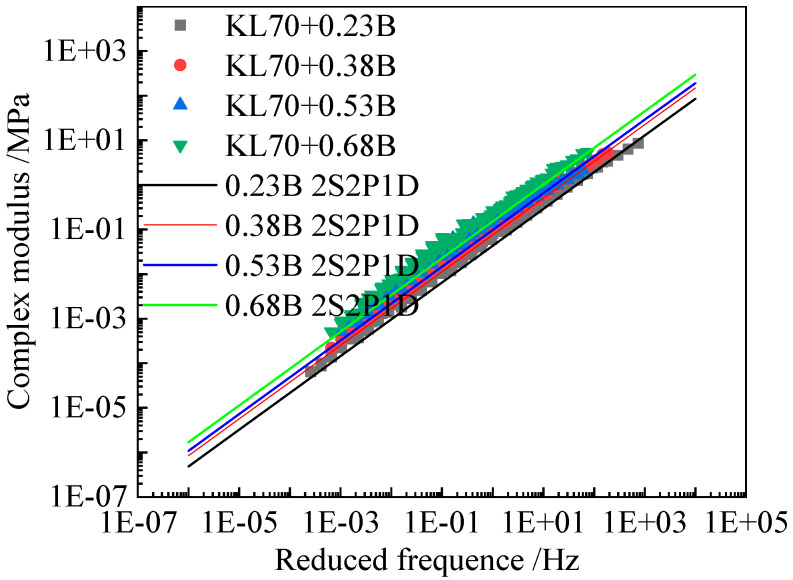
Comparison between the tested complex modulus master curves and 2S2P1D model fitted results of KL70 (B) asphalt mastics.

**Figure 10 materials-15-05688-f010:**
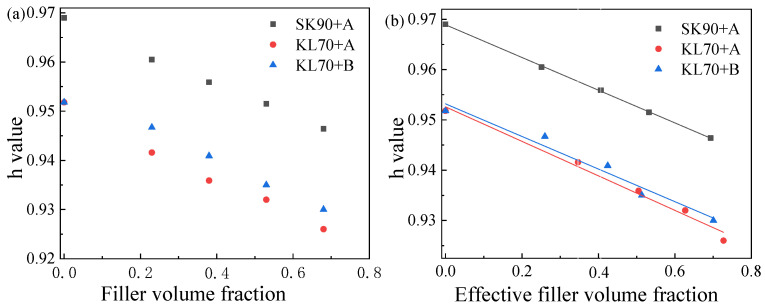
Relationship between h of binders and the corresponding mastics (**a**,**b**). (**b**) is more observed compare with (**a**).

**Figure 11 materials-15-05688-f011:**
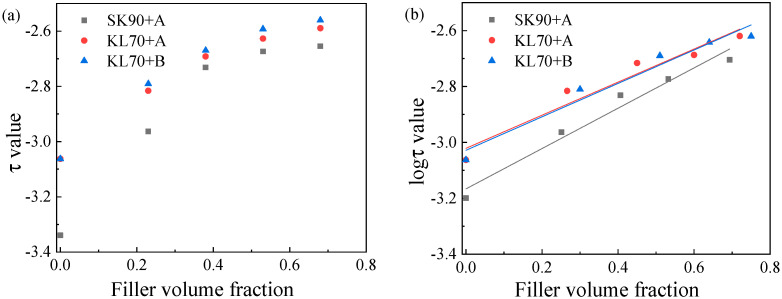
Relationship between log *τ*_0_ of binder and the corresponding mastics (**a**,**b**). (**b**) is more observed compare with (**a**).

**Figure 12 materials-15-05688-f012:**
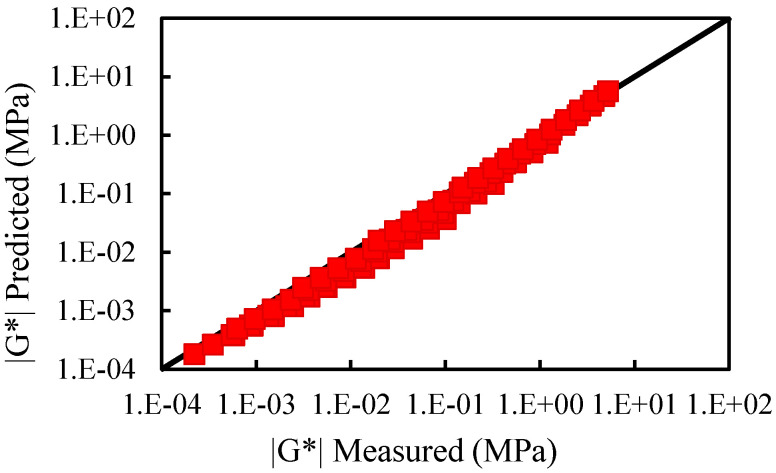
Comparison between the measured complex modulus of ZH70 (0.38A) with the predicted.

**Figure 13 materials-15-05688-f013:**
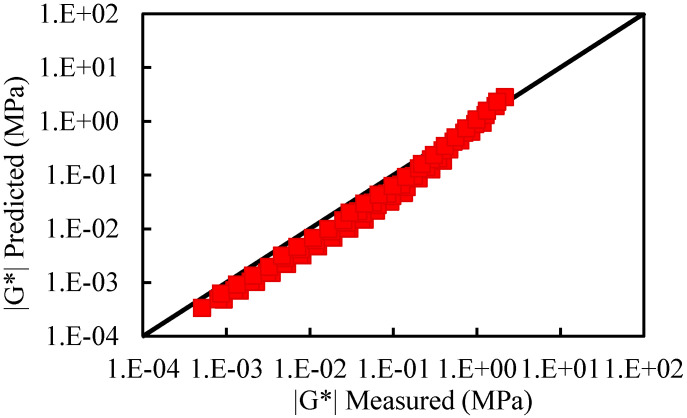
Comparison between the measured complex modulus of ZH70 (0.53A) with the predicted.

**Table 1 materials-15-05688-t001:** Properties of Asphalt Binders.

Asphalt	Penetration 25 °C/0.1 mm	Soft Point/°C	Ductility/cm	PG/°C
SK90	83.0	45.5	85 (10 °C)	52–28
KL70	65.5	48.0	63 (10 °C)	58–22
ZH70	63.0	47.5	50 (10 °C)	58–22

**Table 2 materials-15-05688-t002:** *C* value of different asphalt and filler.

Asphalt Mastic	0.32	0.38	0.53	0.68
SK90 + filler (A)	1.76	1.38	1.12	1.06
KL70 + filler (A)	1.44	1.28	1.08	1.02
KL70 + filler (B)	1.29	1.25	1.07	1.04

**Table 3 materials-15-05688-t003:** Parameters of the 2S2P1D model for all the binders and corresponding mastics.

Asphalt	Mastic	δ	k	h	β	log(τ_0_)	G_0_ (MPa)	G0 (MPa)
SK90	0	10	0.9649	0.9649	1.83 × 10^8^	−3.3392	0	1000
	0.23A	10	0.9615	0.9615	1.83 × 10^8^	−2.9635	0.61 × 10^−6^	1290
	0.38A	10	0.9559	0.9559	1.83 × 10^8^	−2.6916	1.39 × 10^−6^	1540
	0.53A	10	0.9515	0.9515	1.83 × 10^8^	−2.6736	2.50 × 10^−6^	1660
	0.68A	10	0.9464	0.9464	1.83 × 10^8^	−2.6547	5.69 × 10^−6^	1840
KL70	0	10	0.9518	0.9518	1.83 × 10^8^	−3.0624	0	1000
	0.23A	10	0.9416	0.9416	1.83 × 10^8^	−2.8158	2.59 × 10^−6^	1420
	0.38A	10	0.9359	0.9359	1.83 × 10^8^	−2.6916	4.19 × 10^−6^	1690
	0.53A	10	0.9340	0.9340	1.83 × 10^8^	−2.627	9.40 × 10^−6^	1780
	0.68A	10	0.9311	0.9311	1.83 × 10^8^	−2.4894	14.70 × 10^−6^	2000
KL70	0	10	0.9518	0.9519	1.83 × 10^8^	−3.0624	0	1000
	0.23B	10	0.9467	0.9467	1.83 × 10^8^	−2.8310	2.54 × 10^−6^	1490
	0.38B	10	0.9409	0.9409	1.83 × 10^8^	−2.6701	5.04 × 10^−6^	1680
	0.53B	10	0.9350	0.9350	1.83 × 10^8^	−2.5925	11.44 × 10^−6^	1800
	0.68B	10	0.9339	0.9339	1.83 × 10^8^	−2.4603	15.30 × 10^−6^	1920

**Table 4 materials-15-05688-t004:** Fitting result of the relation between *h* and effective filler volume fraction.

Function	*y* = *y*_0_ + *bx*		
Asphalt	SK90 + A	KL70 + A	KL70 + B
*y* _0_	0.9689 ± 0.0010	0.9523 ± 0.0012	0.9532 ± 0.0016
*b*	−0.0325 ± 3.7584 × 10^−4^	−0.0331 ± 0.0023	−0.0324 ± 0.0035
Sum squared residual	1.2003 × 10^−7^	5.3941 × 10^−6^	1.0441 × 10^−5^
*R* ^2^	0.9996	0.986	0.965
$R_{\mathrm{Adjusted}}^{2}$	0.9994	0.981	0.954

**Table 5 materials-15-05688-t005:** Fitting result of the relation between logτ and effective filler volume fraction.

Function	*y* = *y*_0_ + *bx*		
Asphalt	SK90 + A	KL70 + A	KL70 + B
*y* _0_	−3.16626 ± 0.03367	−3.02115 ± 0.03945	−3.02811 ± 0.03676
*b*	0.62132 ± 0.07555	0.59193 ± 0.08219	0.59784 ± 0.07148
Reduced Chi-Sqr	0.00485	0.00654	0.00543
*R* ^2^	0.9681	0.9722	0.9792
$R_{\mathrm{Adjusted}}^{2}$	0.9575	0.96532	0.9688

## Data Availability

Not applicable.

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
