# Peer review of "Predicting Dynamic Properties of Asphalt Mastic Considering Asphalt–Filler Interaction Based on 2S2P1D Model"

_materials, 2022, doi:10.3390/ma15165688_

Round 1
Reviewer 1 Report
The current manuscript investigates the relation between the rheological behavior of asphalt binders and corresponding asphalt mastics. The authors use the Palierne model to assess interaction between asphalt binders and fillers and to determine the effective filler volume fraction. The rheological properties of the binders and mastics are modelled with the 2S2P1D model, and relations between 2S2P1D parameters and the effective filler volume fraction are analyzed. Consequently, the authors present a method to estimate the rheological properties of asphalt mastics.
The topic of the present study is certainly of interest to asphalt researchers and engineers. The state-of-the-art in predicting the rheological behavior of asphalt mastics is reviewed in the Introduction section, and also the characteristics of the 2S2P1D model are sufficiently described. The authors clearly state that the effect of bitumen-filler interaction on the rheological behavior of asphalt mastics is still not fully understood, and they try to address this knowledge gap in the present study. However, I still have several comments to the manuscript:
- There are several instances where the authors refer to other studies without providing any references. For example, on lines 46-50 (page 2), lines 251-253 (page 7) and lines 348-349 (page 10).
- In Section 4.3.2. it is concluded that the variation of 2S2P1D parameters with filler volume fraction does not depend on the type of asphalt binder or filler. However, as the authors highlight in the Introduction section, the rheological properties of asphalt mastics are expected to be significantly dependent on bitumen-filler interaction. The authors should discuss this surprising finding and explain why the experimental data does not show significant dependence on bitumen and filler type.
- The authors highlight in the Introduction section that the addition of filler is expected to change the phase angle of asphalt binder. However, this does not seem to be the case with the asphalt mastics investigated in this study. Please discuss/explain this surprising observation.
- Recent studies employing the so-called 4-mm DSR measurement technique have shown that the glassy shear modulus of unmodified asphalt binders is around 1 GPa. This value should be used instead of the current value of 100 MPa.
- The authors should explain where the elastic modulus and Poisson's ratio of limestone was obtained.
- In Section "2. Materials and Tests" the authors state several times that four asphalt binders were investigated. This should be corrected to the actual number of three binders.
- Table 1: "Duration" should be corrected to "Ductility". Also, please double-check that the PG of the ZH70 binder is correct (the high PG seems unrealistically high for me).
- I am not sure what "Palierne C" refers to, does it refer to the parameter C or to the model itself. Please elaborate.
- Please explain where did you get the H(omega) value of 0.5 for mineral filler.
- Line 342: "shear rate" should probably be corrected to "frequency".
- The term "pulsation frequency" is used in conjunction with the 2S2P1D model. However, to be consistent with the terminology, I would us the term "frequency" or "angular frequency" here.
- Please double-check figure and table numbers, there seems to be a couple of errors.
- Lines 377-379: I do not see how decrease in log(tau0) directly indicates reduction in the temperature dependence of the asphalt binder, please elaborate.
- Table 2: It is not realistic to expect that the values of k and h parameters are identical. There needs to be a better approach to estimate the value of the k parameter. Also, I would expect the value of the beta parameter to be dependent on the type of asphalt binder and filler volume fraction.
- Figures 7-9: I am surprised that the 2S2P1D model fits show a linear dependence of G* on the frequency over the whole frequency range; please double-check if these model fits have been plotted correctly.
- The manuscript contains a lot of grammatical errors and typos. Please perform proper language editing on the manuscript.
Reviewer 2 Report
Thank you very much for this intersting contribution. Please consider following advices:
- regarding content:
- in section 2, sometoimes four tested binders are mentioned (lines 140, 146, 209). Please correct.
- Please check densities and specific surfaces (lines 147-149). It seems not plausible.
- Please add some details about the procedure applied for time-temperature superposition
- results discussed in lines 222-233 could be clearer displayed as a table. would be easier to follow.
- editorial items
- table 1: "ductility instead of duration?
- check table numbers
- line 476 "fraction" instead of "faction"
Reviewer 3 Report
The abstract should be corrected. Describe specifically what was done in the article and what it resulted from. Delete the initial sentences. They are not necessary.
In the literature review, describe other works containing rheological models, e.g .:
Szydło Antoni, Mackiewicz Piotr: 2003, "Verification of bituminous mixtures' rheological parameters through rutting test". Road Materials and Pavement Design, Vol. 4, No. 4, pp. 423-438.
Chiara Riccardi, Augusto Cannone Falchetto, Massimo Losa, Michael P. Wistuba, Rheological modeling of asphalt binder and asphalt mortar containing recycled asphalt material, December 2015, Materials and Structures 49(10), DOI: 10.1617/s11527-015-0779-z
Please describe which models are effective in the description of asphalts and asphalt mixtures.
Better to describe why the study was chosen: dynamic shear rheometer.
Note that other researchers have used static models, and here is a dynamic model.
Round 2
Reviewer 1 Report
I appreciate the time and effort the authors have used to respond to my comments and to revise the manuscript. However, in my opinion, there are still several parts of the manuscript that need to be improved before the manuscript can be potentially approved for publication. My comments are provided in the attached Word file (marked in red). Please remember to not only respond to my comments in the Word file, but also revise the manuscript accordingly when needed.

Author Response
Dear reviewer,
Thank you very much for your good comments and suggestions concerning our manuscript. Those comments are all valuable and very helpful for revising and improving our paper. We have studied the comments carefully and modified the manuscript accordingly. We hope they can meet your approval. The detailed corrections are listed as the word.
